# MicroRNAs: An Update of Applications in Forensic Science

**DOI:** 10.3390/diagnostics11010032

**Published:** 2020-12-26

**Authors:** Anna Rocchi, Enrica Chiti, Aniello Maiese, Emanuela Turillazzi, Isabella Spinetti

**Affiliations:** Institute of Legal Medicine, Department of Surgical Pathology, Medical, Molecular and Critical Area, University of Pisa, Via Roma 55, 56126 Pisa, Italy; enricachiti@gmail.com (E.C.); aniello.maiese@unipi.it (A.M.); emanuela.turillazzi@unipi.it (E.T.); isabella.spinetti@unipi.it (I.S.)

**Keywords:** microRNAs, biomarkers, body fluids, wound vitality, drowning, monozygotic twins, time of death, anti-doping, sepsis

## Abstract

MicroRNAs (miRNAs) are a class of non-coding RNAs containing 18–24 nucleotides that are involved in the regulation of many biochemical mechanisms in the human body. The level of miRNAs in body fluids and tissues increases because of altered pathophysiological mechanisms, thus they are employed as biomarkers for various diseases and conditions. In recent years, miRNAs obtained a great interest in many fields of forensic medicine given their stability and specificity. Several specific miRNAs have been studied in body fluid identification, in wound vitality in time of death determination, in drowning, in the anti-doping field, and other forensic fields. However, the major problems are (1) lack of universal protocols for diagnostic expression testing and (2) low reproducibility of independent studies. This review is an update on the application of these molecular markers in forensic biology.

## 1. Introduction

In recent years, RNA profiling has undergone enormous development in various fields of forensic science, such as identification of body fluid, wound age determination, and post-mortem interval (PMI) assessment [1,2,3]. MicroRNAs (miRNAs) are promising biomarkers in forensic practice owing to their short size, and are particularly valuable for degraded samples or complex mixtures [4,5].

MiRNAs are small, single-stranded, non-coding RNAs of 18 to 24 nucleotides in length, well conserved in eukaryotic organisms. They act at the post-transcriptional level and regulate the expression of many genes in various biological processes, binding complementary sequences of mRNA target and silencing them via degradation through mRNA cleavage or by preventing protein synthesis. Only few bases are necessary for mRNA-miRNA interaction so that a miRNA can have many targets. Imperfect base-pairing results in repression of mRNA translation, while perfect base pairing induces mRNA cleavage. MiRNAs originate from RNA transcripts regions that fold back on themselves to create short hairpins and are synthesized from primary miRNAs (pri-miRNAs) which are transcribed in the nucleus. Primary miRNAs, about 70 nucleotides in length, may contain up to six miRNA precursors. They are converted into pre-miRNAs by Drosha and then carried into the cytoplasm by exportin 5. Pre-miRNAs are cleaved by the RNase III enzyme in duplex miRNAs of 22 nucleotides in length. The guide strand is incorporated into RISC (RNA-induced silencing complex), which is composed of the mature miRNA and by members of Argonaute proteins, while the other strand is being normally degraded. Mature miRNAs incorporated in RISC complexes are less susceptible to degradation processes [6,7].

MiRbase, the bioinformatic database of miRNA sequences, reports that actually 1917 miRNAs are encoded by the human genome [www.mirbase.org]. According to the standard nomenclature system each miRNA is named with the prefix “miR” followed by a dash and a number related to the discovery order.

The first miRNA has been described in 1993 by Lee et al. in *Caenorhabditis elegans*. They isolated a non-coding RNA (*lin-4*) which controlled *C. elegans* larval developmental stages by lin-14 gene repression [8]. Since the 2000s, *lin-7* miRNA and other miRNAs were discovered which were identified as a distinct category of biological regulators [9,10,11,12,13].

Most miRNAs are located within the cell, but some of them have also been found in various body fluids including plasma, tears, cerebrospinal fluid, and saliva, and are commonly being referred to as circulating miRNAs [14,15]. These circulating miRNAs are a particularly promising class of biomarkers given several characteristics they share: first, unlike other RNA classes, they are highly stable even at extreme conditions of temperature, pH, and chemical treatments; and they are preserved from endogenous RNase-activity as they are carried by lipoprotein complexes and RNA-binding proteins in extracellular vesicles [16,17].

miRNA analysis can be performed with profiling methods such as microarray or next generation sequencing (NGS) techniques that allow simultaneous detection of even hundreds of low copy number miRNAs in a single experiment [18].

When few selected miRNAs are analyzed the assay can be performed by quantitative Real-Time PCR (qRT-PCR) which shows an increased sensitivity and reproducibility.

To reduce technical variations introduced during the experimental procedure, target miRNA expression is normalized to a stable internal control, measured in the same sample at the same time. An ideal internal control should be expressed at a constant level, not dependent on biological variations, with length and expression range similar to the target miRNA.

Since miRNAs are differentially expressed in diseased tissues, they have been investigated as potential signatures and biomarkers in several diseases, including cancer, cardiovascular disorders, muscular disorders, and diabetes [19].

In recent years miRNA profiling has been studied in various fields of forensic science. In fact, due to their low molecular weight, their abundant and tissue-specific expression, and their role in regulating physiological and pathological processes, miRNAs carry a great potential in forensic diagnostics [20], although they are still in the early discovery stage.

In this review, we explore the forensic application of miRNAs investigation as a very promising tool for forensic analysis.

## 2. miRNA and Body Fluid Identification

Identification of the biological material found at a crime scene is an indispensable tool in offence reconstruction. Determining whether a specific body fluid is present, and subsequently identifying it, is the first step in forensic investigations usually followed by DNA profiling [21]. Many forensically relevant body fluids can be found at the crime scene such as blood, saliva, semen, vaginal secretions and urine. Current presumptive and confirmatory tests available for forensic body fluid identification (BFID) are rapid and exhibit variable degrees of sensitivity and specificity [21]. However, the destructive nature of some of these screening tests is a crucial point to be considered especially in case of small amount of evidence material. Furthermore, some relevant forensic body fluids (menstrual blood and vaginal material) still lack accepted specific identification tests [22].

DNA-methylation assays, identifying tissue-specific DNA methylation profiles, have been suggested as useful tools to determine the tissue type of a human biological trace [23,24,25,26,27,28,29]. However, such assays exhibit low specificity [30].

Additionally, mRNA seemed to be a suitable candidate for forensic BFID as every cell has its specific gene expression. However, mRNA analysis showed several limitations because mRNA is susceptible to degradation due to the action of cellular ribonucleases and environmental factors (e.g., heat, UV light or humidity), circumstances frequently encountered in forensic caseworks [31].

In the last ten years, miRNAs have been proposed as biomarkers for cell types identification. These biomolecules due to their small size are less susceptible to degradation compared to mRNA (20–25 bases in length versus 200–300 of mRNA) [4]. Moreover miRNAs, as well as mRNAs, can be co-extracted with DNA using common DNA extraction methods. This allows the parallel analysis of BFID test and DNA profiling, very helpful in case of degraded samples or a small amount of forensic traces [32,33,34].

Since numerous studies revealed that some miRNAs were expressed in a tissue-specific manner, at first, forensic researchers aimed to identify miRNAs specific for one body fluid (present in one body fluid but completely absent in the other body fluids). However, the specificity of most miRNAs did not meet these requirements.

Therefore, most of the reports selected candidate miRNAs as those miRNAs found in higher abundance in a specific body fluid (compare to an internal control) but present with lower abundance in other body fluids.

In the last decade, a lot of research has been carried out to identify body fluid origin using miRNAs profiling [33,34,35,36,37,38,39,40,41,42,43,44,45,46,47,48,49,50,51,52,53,54,55,56,57,58].

Microarray platforms highlighted many differentially expressed miRNAs [35,36,37,38] but qRT PCR is necessary to validate data [35,36,37,38,39,40,41,42,43,44,45,46]. qRT-PCR is the common strategy for miRNA profiling determining the relative differences in expression of various miRNAs. In qRT-PCR assays appropriate internal controls are fundamental for data normalization. Some studies evaluated miRNAs as suitable internal controls [42,47].

Another technique for miRNAs identification is NGS with Illumina Hi-Seq platform, introduced by Seashols-Williams et al. [47] and evaluated also by Dorum et al. [48].

A fundamental step in BFID analysis is the ability to obtain a sufficient quantity and quality of miRNAs from a sample. Forensic caseworks are often characterized by limited usable amounts of traces for BFID and thus DNA typing is also constrained, it is difficult to separately extract all targets using different parts of the sample.

Various co-extraction strategies have been evaluated to generate DNA for STR analysis and miRNA for BFID profiles from a single sample [32,33,34,45,46].

Two relevant studies reported the co-analysis of both BFID and DNA profiling using a co-extraction method followed by capillary electrophoresis [49,50].

Li and colleagues developed a four-marker miRNA BFID system based on a set of specific linear RT-PCR primers and one pair of PCR primers (one dye-labeled universal and one specific for each miRNA) for the endpoint PCR. The multiple system could discriminate venous blood, menstrual blood, and semen [49].

Mayes et al. designed an eight marker multiplex system for BFID (including let-7g as internal control) to differentiate venous blood, menstrual blood, semen, and saliva. They introduced additional primers and dyes in the PCR assay [50], using the same approach of Li et al. [49]. All samples tested in both studies yielded full STR profiles [49,50].

The intention of this method was to select specific markers present in high abundance to reduce amplification of background transcription. However, the complexity of appropriate primers design and markers cross reactivity represent important limits for this attractive approach.

Another crucial point in miRNAs analysis is the use of appropriate statistical methods to determine whether there is a statistically significant difference between the two observed data. Several studies used one-way analysis of variance (ANOVA) and independent two-sample t-tests [36,48,51].

He et al. developed a mathematical model based on the Fisher discriminant function to differentiate between five relevant forensic body fluids, with a validation accuracy of the identification of 100% [43,44].

In the first study, the authors developed a stepwise strategy to distinguish menstrual from venous blood using only four miRNAs selected from the literature [43].

In the second study, the analysis was extended to nine miRNAs allowing the discrimination of venous blood, menstrual blood, saliva, semen and vaginal secretion. The self-validation accuracy of the model was 99.7% [44]. Results were validated in a large number of samples (40–70 samples for each body fluid) [43,44].

The first study on miRNAs and forensic BFID was conducted in 2009 by Hanson et al. The authors examined 452 miRNAs in five forensic relevant body fluids (semen, venous blood, saliva, menstrual blood, and vaginal secretion) with qRT-PCR. They identified a panel of nine differentially expressed miRNAs with a high degree of specificity in each body fluids. In particular, miR-451a and miR-16 were higher in venous blood; miR-412 and miR-451a were identified in menstrual blood, miR-135b and miR-10b in semen; miR-205 and miR-658 in saliva while miR-124a and miR-372 were identified in vaginal secretions. Furthermore, the small quantity of total RNA (50 picograms) needed for miRNA analysis underlined the usefulness of these markers in case of a low amount of forensic samples [39].

One year later, Zubakov et al. investigated a set of 718 miRNAs in five body fluids with microarray. Two miRNAs for venous blood (miR-144 and miR-185) and two for semen (miR-135a and miR-891a) were confirmed as differentially expressed with qRT-PCR, suggestive to be useful for venous blood and semen identification. Moreover, these four miRNAs were not degraded in samples stored for 1 year and could be revealed starting from few picograms of total RNA [35].

Courts and Madea performed an array that examined 800 miRNAs in venous blood and saliva and validated six candidate miRNAs. They proposed miR-126, miR-150, miR-451a for the identification of venous blood and miR-200c, miR-203, miR-205 for saliva [36].

Screening a total of 754 miRNAs, Wang et al. proposed five body fluid-specific miRNAs, whose detection were highly sensitive: miR-16 and miR-486 for venous blood, miR-214 for menstrual blood, and miR-888 and miR-891a for semen. They highlighted three new miRNA markers (miR-486, miR-888, and miR-214) but did not identify specific miRNAs for saliva and vaginal secretions [37].

Two years later, the same authors aiming to find a strategy for saliva identification, analyzed eight potential saliva miRNAs selected from previous reports [36,37,39] in five body fluids. Only three miRNAs (miR-200c-3p, miR-203a, miR-205-5p) showed different expression patterns among these body fluids. In particular, miR-200c-3p was higher in vaginal secretions compared to other body fluids; miR-203a was higher in menstrual blood, vaginal secretions, and saliva and not present in venous blood; miR-205-5p was higher in menstrual blood and vaginal secretions compared with saliva and semen, but not detected in venous blood. None of the investigated miRNAs could independently differentiate saliva from other body fluids. Then the authors developed a stepwise strategy using miR-200c-3p, miR-203a or miR-205-5p in combination with previously characterized body fluid-specific miRNAs. For example, miR-200c-3p in combination with miR-214-3p (highly expressed in menstrual blood) and miR-486-5p (highly expressed in venous blood) could separate saliva from blood while miR-205-5p in combination with miR-891a (a semen-specific miRNA) was able to separate saliva from semen [40].

Luo et al. performed a genome-wide profiling using an array containing 3100 capture probes covering all human miRNAs. They selected 10 relevant miRNAs, eight novel miRNAs and two previously reported for semen identification (miR-135a-5p and miR-888-5p) but they didn’t perform qRT-PCR to validate data [38].

A year later Sauer et al. published a comprehensive study based on a thoroughly validated qRT-PCR procedure. Four miRNAs (miR-144-3p, miR-891a-5p, miR-203a-3p and miR-891a-5p) were reported useful for BFID but only miR-891a-5p was confirmed as truly semen specific. The authors proposed a decision algorithm to detect each body fluids employing few markers to simplify the analysis: miR-891a-5p for semen identification, miR-144-3p to discriminate blood from non-blood samples, miR-144-3p and miR-203a-3p to distinguish between venous and menstrual blood, miR-203a-3p and miR-124-3p to differentiate between saliva and vaginal secretions [41].

In a study conducted on 15 miRNAs (selected from the literature) in six body fluid samples (including skin), Sirker et al. validated four candidate internal controls (miR26b, miR92, miR144, and miR484) as the most stably expressed across the samples analyzed [42]. Normalizing qRT-PCR data with these controls revealed that only five miRNAs (miR-10b, miR-203, miR-374, miR-451a, and miR-943) were able to differentiate between six different cell types. miR-451a was a strong biomarker for all blood types identification. The combination of miR-451a and miR-943 could distinguish blood from other body fluids, while miR-943 was useful to separate menstrual blood from skin. The combination of miR-10b, miR-203, and miR-943 allowed distinguishing between venous and menstrual blood, while miR-10b and miR-451a distinguish skin from saliva. miR-374 alone could discriminate semen from blood samples. miR-203 could separate vaginal secretion from saliva, semen, and blood [42].

Seashols-Williams evaluated other forensically relevant body fluids: urine, feces, and perspiration [47]. The authors identified six candidate miRNAs (miR-200b, miR-1246, miR-320c, miR-10b-5p miR-26b and miR-891a) and two suitable internal controls (let-7g and let-7i) for data normalization. They proposed a miRNA panel for BFID: miR-200b could distinguish venous and menstrual blood from the other body fluids while miR-1246 could separate venous from menstrual blood. MiR-10b-5p differentiated urine and feces from other body fluids while miR-320c distinguished urine from feces, miR-26b was higher in saliva than in other body fluids. They also found miR-891a as an indicator for semen [47].

In the last decade various miRNAs have been suggested as candidate biomarkers for forensic BFID but several discrepancies arise between published reports concerning identified miRNAs. Only some miRNAs overlap between studies (Table 1).

The reason of these inconsistencies could be attributed to differences in platforms, detection chemistries, and normalization to different internal controls. For example, qRT-PCR is more sensitive in identifying the miRNAs of interest than NGS analysis.

Table 1 reports miRNAs identified in the literature as potential biomarkers for BFID in five forensic relevant body fluids.

As shown in Table 1, miR-451a has been reported a strong candidate biomarker for blood identification [33,36,39,42,43,44,49,50,58] able to discriminate blood from non-blood samples; also miR-144-3p [35,41,44,58] and miR-16 [37,39,58] seems to be useful to separate venous blood from other body fluids.

miR-214-3p [37,43,44,49] and miR-412 [39,50,52] are a suggested candidate marker for menstrual blood because their expression levels are higher in menstrual blood compared to other body fluid. miR-203a [36,39,41,58] and miR-205-5p [33,36,39,44,50] are suggested as a candidate marker for saliva.

Different studies confirmed miR-891a [35,37,41,44,47,50,53,54,58], miR-10b [39,42,50] and miR-888 [37,38,44,49,53,54,58] as semen specific markers, especially miR-891a peculiar of the epididymis tissue [53,54]. miR-203a [40,41,42] was instead suggested for vaginal secretion identification.

An ideal BFID test should be highly specific, easy to perform and should require a low quantity of starting sample allowing the recovery of miRNAs also in minute traces.

Currently, there are few strong candidate miRNAs with high specificity for one particular body fluid. While semen and blood in general can be determined with high certainty, differentiation of venous and menstrual blood (as well as saliva and vaginal secretion) remained most challenging. Owing to a lack of highly-specific markers the only reliable method to improve the specificity for forensic BFID is a step-by-step strategy that combines multiple specific markers.

However, this method is suitable for identifying the type of body fluid in samples from a single source of body fluid and it is not appropriate for mixed samples.

## 3. miRNAs and Wound Vitality

Distinguishing vital reactions from post-mortem changes and establishing wound age are relevant issues in forensic autopsies [59,60]. These phenomena are strictly intertwined, in particular regarding skin lesions, which represent one of the principal fields of research in forensic medicine [61,62].

When the skin is injured wound healing process starts. This process is well regulated and is divided into four overlapping phases: hemostasis, inflammatory, proliferative, and remodeling phase. Each stage is characterized by specific morphological and/or biochemical features [63].

Over the years, immunohistochemical markers, adhesion molecules, cytokines and chemokines, growth factors, and wound healing-related molecules have been evaluated for wound vitality determination [64]. Additionally, miRNAs have been studied as possible biomarkers for this issue because previously published reports demonstrated a key role of specific miRNAs in the regulation of the wound-healing process [65,66,67,68,69,70,71,72,73,74,75].

In particular: miR-21, miR-146a/b miR-142 and miR-155 are characteristic of the inflammatory phase [65,66,67,70], miR-21 [68,69] and miR-132 [71] are involved both in inflammatory and proliferative phase, miR-31 [72] and miR-99 [73] are characteristic of the proliferative phase while miR-29a,b,c and miR-192 are involved in remodelling phase [75,76].

To the best of our knowledge, very few studies analyzed miRNA differences between ante and post-mortem skin lesions [77,78].

A recent investigation adopting an animal model determined wound vitality in incisional wounds. In this study, 18 rats were sacrificed after a full-thickness skin incision and skin samples were assessed immediately after death (I group), 24 h after death (II group), and 48 h after death (III group). Histological examination revealed an early inflammatory reaction in the wound area after incision (I group) and with increasing time presence of collagen fibrous (II group) and hemosiderin depositions (III group). miRNA analysis showed significantly higher expression levels of miRNA-21 and miRNA-205 in skin samples assessed 24 h after death than in the other groups. miRNA-21 and miRNA-205 were continuously expressed up to two days after death but at lower levels. No significant negative correlation was observed between the time passed after death and miRNA expression [77].

In another recent study, Neri et al. investigated miRNA expression changes in the skin ligature marks of subjects who died by hanging. The authors found increased levels of miR125a-5p and miR125b-5p, two miRNAs involved in the inflammatory phase of wound healing, and also increased levels of miR92a-3p, miR128-3p, miR130a-3p (miRNAs with an anti-inflammatory role) compared to control skin samples [78].

### Exposition to Fire

Another very interesting field of forensic investigation is the question of whether a victim was exposed to the fire before or after death.

The diagnosis of vitality regarding burned bodies is still fraught with problems [79]. Mainly due to the extensive damage that fire can cause to a body, differentiation between ante-and post-mortem injuries may be challenging and detection of signs of vitality may be resolutive [59].

MiRNAs regulate the expression of genes related to the wound-healing process, including the genes involved in thermal burn injury [80].

There is only one recent paper on miRNAs as a potential marker for the identification of ante mortem burns and postmortem burns conducted on an experimental burn model. The authors performed a microarray analysis to identify differentially expressed miRNAs in the skin of an experimental burned mice model [81]. Twenty-four miRNAs were significantly deregulated in ante-mortem burned mice skin: 19 miRNAs (miR-135a-1-3p, miR-183-3p, miR-188-5p, miR-3081-5p, miR-5103, miR-6378, miR-6385, miR-6391, miR-6769b-5p, miR-6969-5p, miR-7005-5p, miR-7036a-5p, miR-7044-5p, miR-710, miR-711, miR-7118-5p, miR-7668-3p, miR-8090, miR-874-3p) were upregulated and five miRNAs (miR-155-5p, miR-28a-3p, miR-467b-3p, miR-5132-5p, miR-6924-3p) were downregulated if compared to unburned mice skin. The results of microarray analysis were confirmed by RT-qPCR analysis conducted on 10 of the 24 miRNAs randomly selected. No differences were observed between post-mortem burned skins and unburned specimens indicating that these miRNAs were not affected by postmortem burn [81]. This study suggests that the investigated miRNAs could be considered promising biomarkers for vital reaction to burns, although further investigations also on human samples are required to confirm the author’s results.

## 4. miRNAs and Time of Death Determination

Estimation of post-mortem interval (PMI), the time interval between death and the examination of the deceased, is one of the most challenging forensic questions. Assessment of post-mortem physical changes of the body and entomological approaches are useful to evaluate PMI respectively within the first 24 h and at a more advanced stage. These routine procedures are not very accurate and are influenced by environmental conditions and individual characteristics.

Different methods were developed to define a more precise PMI estimation such as molecular methods based on the time-dependent degradation of proteins, DNA, and RNA.

However, post-mortem RNA degradation by ribonucleases and bacteria is influenced by other factors, especially environmental conditions (temperature, sunlight, humidity) and by cause and circumstances of death [82]. RNA post-mortem stability is also affected by the type of tissue considered: brain, heart and skeletal muscle are more conserved compared to pancreas and liver (ribonucleases rich organs) [83].

Among the RNA species, miRNAs are potentially valuable in estimating PMI in advanced stage for their numerous characteristics, especially stability. In fact, miRNAs are less susceptible to degradation respect to mRNA because of their small size and protection by protein and/or lipid matrices. To note, also miRNAs stability is influenced by factors as the type of tissue, environmental conditions (high temperatures) or body condition as putrefaction.

Several recent works reported the recovery of some miRNAs after compromising treatments and challenging environmental conditions [16,17,84].

let-7 g, let-7i, miR-200b-5p, and miR-891a-5p resisted to UV treatment in blood, urine, semen, and saliva [16], while miR-451a and let-7g were recovered in blood and semen stains aged six months exposed to different experimental conditions: sunlight, humidity (99%) and heat (60%) [17]. These miRNAs persisted in laundered stains [17]. Minimal degradation of others miRNAs (including miR-451a) in bloodstains aged five months at 37 °C was observed also by Fang et al. [84].

Several studies reported no correlation between miRNA expression levels and early PMI (within 24 h) [85,86,87].

Li et al. published the first report regarding miRNAs and PMI determination [85]. Researchers evaluated the expression levels of two RNA markers (miR-1-2 and 18S rRNA) in rats’ cardiac tissues at different PMI in a controlled temperature system (25 °C). 18S rRNA levels increased within 96 h after death and then declined gradually. miR-1-2 levels were fairly stable within 120 h but started declining after this period. The authors concluded that miR-1-2 and 18S rRNA could be useful PMI biomarkers within 96–120 h after death [85].

In a further study, Wang et al. analyzed the expression levels of other miRNAs (miR-122, miR-133a, miR-150, miR-195, miR-206) within 48 h after death in mice liver tissues. They found no correlation between miRNAs expression levels and early PMI. miR-206 and miR-133 levels decreased 24 h after death suggesting a possible role for PMI determination between 24 and 48 h [86].

A recent study performed by Na et al. highlighted the potential of miRNAs for PMI estimation in advanced body decomposition [87]. The authors investigated the expression levels of let-7e and miR-16 in 71 human bones (patella) [87]. The samples were divided into four PMI groups (PMI < 1 month, 1 month < PMI < 3 months, 3 months < PMI < 6 months, PMI > 6 months). Results reported that let-7e and miR-16 levels decreased with PMI increasing and were significantly different between the first and the other three PMI groups. The authors suggested that let-7e and miR-16 could be candidate markers to estimate PMI of several months in bone tissue [87].

### 4.1. miRNAs as Circadian Rhythm Biomarkers

Few publications investigated miRNAs that regulate clock genes to determine the time of the day at death, evaluating miRNAs expression levels during day and night [88,89,90].

Odriozola et al. performed an array in 34 human vitreous humor samples, a well preserved tissue after death [88]. They confirmed two miRNAs (mir-142-5p and mir-541) as differentially expressed between day and night-time. In particular, mir-142-5p and mir-541 levels were lower in daytime deaths and higher in night-time deaths. Besides these miRNAs were stable at least for 24 h after death [88].

Moreover, in another study miR-142-5p and miR-541were analyzed in blood, the forensic trace most encountered at crime scenes, but they were not useful to estimate the deposition timing of this body fluid [89].

Corradini et al. analyzed 10 miRNAs selected from the literature with a supposed role in circadian rhythms in 12 post-mortem vitreous humor and blood samples from subjects who died during the day or night [90]. Four miRNAs showed significant differential expression between individual died at day and at night: miR-106b and miR-96 in vitreous humor (miR-106b and miR-96 levels were higher in daytime deaths compared to night-time deaths) while mir-142-5p and mir-219 in blood (mir-142-5p and mir-219 levels were higher in daytime deaths compared to night-time deaths) [90]. However, the authors couldn’t replicate the results of previous work [84] probably due to the smaller size of samples analyzed [90].

### 4.2. miRNAs as Internal Controls in PMI

Since miRNAs exhibit high stability, many studies evaluated these biomolecules as internal controls in qRT-PCR analysis for PMI estimation in an advanced stage [91,92,93,94,95].

Pan and et al. studied five RNA markers (β-actin, GAPDH, 18S rRNA, 5S rRNA, and miR-203) in 18 skin rat samples at different PMIs (0–120 h post-mortem), and three different environmental temperature groups (4 °C, 15 °C, and 35 °C). Results showed that miR-203 and 5S rRNA were more stable at 4 °C and 15 °C compared to the other RNA markers [91].

In the same year, Lv et al. analyzed other RNA markers including miR-125b and miR-143 in 12 rat spleen samples [92]. The rats were divided into two different temperature groups: I Group was kept at 25 °C and samples were taken for up to 144 h of PMI, while II group was kept at 4 °C and samples were taken for up to 312 h of PMI. The authors concluded that mir-125b and miR-143 were the most stable analyzed RNA markers (at 25 °C were stable PMI < 36 h, at 4 °C were stable PMI < 144 h) and suitable in estimating late PMI. Furthermore, PMI determination was much accurate at 25 °C than at 4 °C [92].

Subsequent studies were performed on a larger number of samples [93,94,95].

Lv et al., two years later, studied nine RNA markers in lung and muscle tissues of 216 rats and in 12 autoptic samples with known PMI. miR-195, miR-200c, 5S, U6, and RPS29 were selected for lung while miR-1, miR-206, 5S, and RPS29 for muscle tissues [93]. The samples were stored at different controlled temperatures (10 °C, 20 °C, 30 °C) and were collected for up to 144 h post-mortem. The authors demonstrated that RNA degradation at high temperatures was faster in the lung compared to muscle tissues [93].

Ma et al. analyzed nine RNA markers (β-actin, GAPDH, RPS29, 5S rRNA, 18S rRNA, U6 snRNA, miR-9, miR-125b, and Let-7a) in 270 rats brain samples. The samples were collected up to 144 h of PMI and were divided into four different temperature groups (4, 15, 25 and 35 °C). miR-9 and mir-125b showed higher stability compared to the other RNA markers at the above-mentioned temperatures up to 144 h post-mortem. Ma et al. also created a mathematical model for PMI determination in rat brain tissue using RNA degradation pattern at different temperatures (4, 15, 25 and 35 °C) [94].

The same authors conducted a study on different RNA markers (conserved across species in rat and human tissues) to screen useful candidates and to validate the corresponding mathematical model in humans [95]. Researchers found that miR-1 and miR-133a, together with 5S, were stable over five or more days (also at 35 °C) and, for this reason, they were selected as internal controls. The mathematical model that they validated showed high predictive power, with an estimated error of 5.06 h between real and estimated PMI [95].

Tu et al., evaluating the stability of some RNA markers in mouse myocardium, skeletal muscle, and liver, selected miR-122 and miR-133a in heart, miR-122 in liver, and miR-133a in muscle tissue as internal controls [96].

A more recent study was conducted by the same authors, on murine heart, skeletal muscle, and liver, to develop a mathematical model for PMI estimation. miR-122, miR-133a and 18S, LC-Ogdh, and circ-AFF1 were analyzed as internal control markers for those three tissues, while GAPDH, RPS18, U6, and β-actin were used as candidate biomarkers. It was demonstrated that advanced stages of PMI could be estimated with high accuracy for cardiac tissue. An error rate of 1.5 days in 41% of muscle tissue and 80% of liver tissue was obtained, using the selected reference genes and the target biomarkers mentioned above. Integrating the estimated results of the three tissue and all biomarkers, the accuracy of the PMI determination improved with an estimated error of 0.5 days [97].

Table 2 reports miRNAs evaluated in PMI studies.

Finally, we can conclude that encouraging results come from the evaluation of miRNAs in late PMI (24–144 h after death) [85,86,91,92,93,94,95,96] while no correlation have been found between early PMI (within 24 h) and miRNA expression levels.

In several studies, miR-125b and miR-9 were confirmed as appropriate internal controls compared to other PMI markers in spleen and brain tissues (stable till 144 h after death [92,94,98,99].

It is also necessary to increase the number of studies conducted on human samples.

At present, a multi-method approach (evaluation of algor mortis, livor mortis, rigor mortis, biochemical changes in the vitreous humor, and entomological approach) is recommended for an accurate PMI determination and miRNA profiling could represent an ideal tool to supplement the other classical biochemical and molecular techniques.

## 5. miRNAs and Drowning

Drowning is a fatal event caused by asphyxiation following prolonged immersion in a liquid [100].

Diagnosis of drowning is one of the major challenges in forensic medicine and is usually performed with pathology and laboratory findings combining different techniques, such as evaluation of fluid electrolyte differences in blood, pleural liquid, and vitreous humor of the victims, analysis of aquaporin expression, investigation of diatoms [101,102,103,104,105,106].

Different molecular methods have been employed to find specific biomarkers of drowning such searching of diatoms DNA in victim’s tissues, analysis of intrapulmonary expression of the receptor for advanced glycation end products (RAGE), aquaporin-5 (AQP5), surfactant protein-A (SP-A), interleukin 6 (IL-6) and interleukin 1β (IL-1β) as markers of drowning [103,104,105,106].

In different drowning circumstances, there are changes in body microenvironmental factors such as oxygen saturation and blood electrolytes concentrations because inhaled water can modify water channel regulation or ion transport.

Some miRNAs regulate genes encoding for ion channels [107] and the microenvironmental changes associated with the death process might determine different miRNA expression patterns, and specific miRNAs could be used as reliable biomarkers to confirm the type of drowning.

To the best of our knowledge, only one study, conducted by Yu et al., analyzed miRNA expression profiles in an experimental drowning animal model [108]. Experiments were conducted on 13 mice that were killed as follows: three controls were sacrificed by cervical dislocation, five mice were drowned in freshwater (FW), five mice were drowned in saltwater (SW). Differences between miRNA expression profiles were evaluated in brain samples of SW and FW drowning mice by microarray analysis. A total of 158 miRNAs were found to be differentially expressed but only eight miRNAs (miR-6394, miR-706, miR-30c-1-3p, miR-6238, miR-494-3p, miR-669h-3p, miR-135a-1-3p, miR-5109) were significantly upregulated in FW and downregulated in SW drowning compared to controls. Analyzing the target genes of the differently expressed miRNAs, researchers predicted that the non-selective cation channel HCN1, which is mainly expressed in the mouse brain and cardiac cells, was a miRNA-706 target. The authors noticed that HCN1 mRNA was less expressed in FW and overexpressed in SW samples and suggested miRNA-706 as a candidate biomarker in forensic investigations to discriminate between SW and FW drowning [108]. However, this is only a preliminary study, and further investigations with a larger sample cohort are needed to consolidate the author’s findings.

## 6. miRNAs and Discrimination of Monozygotic Twins

Monozygotic twins (MZ twins) arise from a single cell and they share almost the same genomic DNA sequence. This feature represents a limit in criminal investigations or in paternity cases when the suspect or the alleged father is one of the twins. Differentiating MZ twins in these special cases is a crucial point to confirm the identity and common forensic DNA testing (STR profiling) is not helpful.

In the last years, different approaches were performed to differentiate between MZ twins, such as whole genome sequencing [109] and mitochondrial DNA analysis [110].

Furthermore, the analysis of the epigenetic differences (especially DNA methylation patterns) have been proposed to distinguish MZ twins. However, DNA methylation results in little genomic differences between MZ twins [111,112,113].

Genome screening methods and methylation analysis are costly, time-consuming, and very complex for routine application.

miRNAs have been investigated as biomarkers for MZ twins’ identification because several studies reported some disease-related miRNAs as differentially expressed in MZ twins [114,115,116].

Two recent works analyzed circulating miRNAs expression patterns with the aim of identifying miRNA panels potentially usable to distinguish MZ twins [117,118].

Fang and co-workers performed genome-wide profiling of blood miRNAs from four pairs of healthy MZ twins. Numerous miRNAs were obtained from NGS but only 14% were found differentially expressed in MZ twins (14 miRNAs in pair 1, 26 miRNAs in pair 2, 10 miRNAs in pair 3, 41 miRNA in pair 4). The miRNAs with the most significant differences in expression between the twins were confirmed using qRT-PCR. In this study, only miR-451a showed differential expression within all four pairs of MZ twins [117].

Different results were instead reached by Xiao et al., who analyzed miRNA expression profiles of seven pairs of MZ twins with microarray and detected, on average, 78 differentially expressed miRNAs in each MZ pair. Within these miRNAs, 10 differently expressed miRNAs were selected for further validations via qRT-PCR assays, increasing the sample size to eighteen pairs of MZ twins. The most promising miRNAs for MZ twins’ discrimination in this work resulted miR-151a-3p, miR-3653-3p, and miR-142-3p [118].

Given the deficiency of studies on this forensic topic, it is currently not possible to establish miRNAs biomarkers for MZ twins’ discrimination. Larger cohort studies are needed to validate the above mentioned miRNAs and to discover new potential miRNAs.

## 7. miRNAs and Anti-Doping

Doping is a complex phenomenon that often involves the use of multiple substances to illegally improve athletic performance. Over the years, the anti-doping techniques, and their ability to detect a wide spectrum of drugs have been a great development [119,120].

A critical issue in doping analysis is the identification of sensitive, non-invasive, and specific biomarkers able to easily reveal doping substances in biological samples. Current methods like immunoassay technology and gas chromatography-mass spectrometry (GC-MS) are limited by a short time frame in which detection of the substances is possible.

Researchers hypothesized that the administration of an exogenous drug could alter the expression levels of specific miRNAs in blood. These molecules are highly stable in plasma and can be detected over a longer period; besides are not affected by environmental factors such as improper storage during blood sample transportation [121].

MiRNAs profiling is performed by qRT-PCR, a highly specific and sensitive technique that allows the simultaneous analysis of multiple targets even in case of small amounts of samples [122]. Therefore profiles of circulating miRNAs have been investigated for potential use as novel, non-invasive anti-doping biomarkers [123,124].

The World Anti-Doping Agency (WADA) reports a list of illegal substances but the most investigated in this field are erythropoietin, growth hormone, and testosterone.

The first work regarding miRNAs and anti-doping biomarkers was conducted by Leuenberger et al. in 2011 on six human plasma samples [123]. Researches demonstrated that a single injection of erythropoiesis-stimulating agent (ESA) was associated with increased levels of circulating miR-144, a miRNA that plays an essential role in erythropoiesis in various organisms. MiR-144 levels remained high and stable till 27 days after ESA stimulation [123].

A further study conducted in mice models confirmed these data [124].

Results from human and animal model suggested a possible use of miR-144 as a new biomarker to detect ESA in plasma.

Other research focused on recombinant human growth hormone (rhGH) abuse detection [125,126].

Kelly and et al. reported that patients with growth hormone deficiency treated with therapeutic replacement doses of rhGH showed lower levels of four circulating miRNAs (miR-663, miR-2861, miR-3152, and miR-3185) compared to subjects with naturally high levels of GH and to controls [125].

Conversely, Lehtihet et al. in a recent paper demonstrated that plasma levels of miR-3152 and miR-3185 were not affected by the administration of low doses of rhGH [126].

Discrepancies between the two studies could be attributed to the different samples analyzed (subjects with GH deficiency treated with therapeutic doses in the first study vs. healthy men in the second one).

MiRNAs were also investigated for testosterone (T) abuse detection [127,128]. Testosterone is an endogenous androgenic anabolic steroid but is considered a performance-enhancing drug in the anti-doping field.

Salamin et al. analyzed three potential candidate miRNAs (miR-122, miR-150, and miR-342) in plasma from 19 healthy male volunteers after transdermal testosterone administration. The authors found that miR-122 (a liver-specific miRNA) levels were increased 3.5 times one day after drug intake [127].

Contrasting data were presented in another study in which miR-122 levels did not change after a single application of a gel containing testosterone [128].

The discordance between these two studies may be caused by the use of different control miRNAs or by differences in absorption, distribution, metabolism, and excretion between the two administration methods described.

Furthermore, circulating miRNAs were investigated for autologous blood transfusion (ABT) detection [129,130,131].

It was demonstrated that after ABT, levels of three plasma miRNAs (miR-26b, miR-30b, miR-30c) increased up to three days after transfusion, while erythropoietin levels decreased [129].

A more recent study reported seven erythroid-related miRNAs (miR-92a-3p, miR-126-3p, miR-144-3p, miR-191-3p, miR-197-3p, miR-486-3p and miR-486-5p) to be upregulated after ABT, while haematological parameters showed moderate changes [130].

These results suggest that autologous blood transfusion may determine an increase of some circulating miRNAs, which could be used as anti-doping biomarkers.

ABT is associated with a long period of blood bag storage (a few weeks minimum) before reinfusion. For this reason, Haberberger and et al. [131] evaluated if the non-physiologic storage of blood was related to changes in function and physiology of erythrocytes. They compared miRNA levels in fresh and stored blood to develop a miRNA panel for ABT detection. Among 28 differentially expressed six miRNAs (miR-16-2-3p, miR-1260a, miR-1260b, miR-4443, miR-4695–3 p, and miR-5100) were upregulated after six-week storage while the other 22 miRNAs were upregulated after blood bag processing (filtration, addition of preservative solution, and subsequent centrifugation) [131].

Few studies have been conducted on circulating miRNAs as biomarkers for anti-doping detection and plasma miRNA expression profiles are not always concordant between different studies.

Currently, it is not possible to utilize these biomolecules as doping biomarkers but a possible future use of miRNAs in anti-doping settings is desirable. More research is needed, with larger cohort studies of athletes, in order to find concordant results and construct a specific miRNA signature for different substances.

## 8. miRNAs and Sepsis

Sepsis is a life-threatening condition caused by a dysregulation of the inflammatory response that still determines a high mortality rate [132]. The pathogenesis of sepsis remains poorly understood and also post-mortem diagnosis is still challenging due to the lack of specific histological evidence [133,134]. A complete methodological approach including clinical data, autoptic, and laboratory findings is necessary for assessing sepsis-related deaths [135].

Since they play a key role in inflammatory responses, miRNAs have been proposed as potential sepsis biomarkers [136]. MiRNAs are involved in the innate and adaptive immunity regulating the TNF pathway [137,138] and the TLR/NF-κB signaling pathway [136].

MiRNAs have been suggested to control the expression of sepsis-related genes (i.e., IL-6 and TNF) but it has also been speculated that miRNAs expression is in turn regulated by these factors [139].

A recent study by Reithmair and et al. proposed a set of miRNAs for an accurate and objective assessment of sepsis-related deaths [140]. Researchers analyzed miRNA expression profiles in blood samples of 22 human patients with sepsis and 23 controls. The miRNA expression profiles were established in different compartments: exosomes, serum, and blood cells. The authors found 103 miRNAs with higher levels and 77 miRNAs with lower levels in septic samples compared to controls. Furthermore, results revealed qualitative and quantitative differences in miRNA profiles between compartments in particular exosomes showed a higher number of miRNAs compared to serum and blood cells. Besides miR-27b-3p levels were higher in exosomes and blood cells than in serum. Furthermore, miR-199b-5p was identified as a potential early biomarker to distinguish patients with sepsis from controls [140].

Finally, it has recently been suggested that circulating miRNAs could also be indicators for specific pathogens.

Wu et al., in a study conducted on an animal model, demonstrated a correlation between blood levels of seven miRNAs (miR-122, miR-133a, miR-133b, miR-205, miR-714, miR-1899, and miR-291b) and Gram-positive infection with *Staphylococcus aureus* [141]. Gram-negative sepsis with *Escherichia coli* was instead related with high levels of other eight miRNAs (miR-16, miR-17, miR-20a, miR-26a, miR-26b, miR-106a, miR-106b, and miR-451a) [141].

Specific variations in miRNAs expression were also found in different bacterial infections due to *Helicobacter pylori* [142], *Mycobacterium tuberculosis* [143], *Listeria monocytogenes* [144], *Staphylococcus aureus* [145], *Salmonella enterica* [146], *Pseudomonas aeruginosa* [147,148], and *Brucella melitensis* [149]. Furthermore, parasitic infections with *Leishmania* [150], *Cryptosporidium parvum* [151,152,153], and *Plasmodium vivax* [154] showed different miRNA expression profiles.

Identification of the responsible pathogens, particularly in healthcare-associated infections, could be decisive in sepsis negligence claims.

Research regarding miRNAs as sepsis biomarkers needs to be improved and more studies, with a bigger analysis cohort, are necessary to get a clearer overview of this topic.

## 9. Concluding Remarks

Since their discovery in 1993, miRNAs have been recognized as critical players in almost every biological pathway.

In recent years, miRNA profiling acquired an increasing interest among clinicians and researchers, both in diagnostic and therapeutic fields, and it has been also explored in the forensic field.

Unique features displayed by miRNAs such as their specificity, their ability to be assayed in parallel to DNA, their stability and low susceptibility to degradation could make these biomolecules ideal biomarkers for forensic practice.

A recent report by Glynn [155] evaluated the potential application of miRNA profiling in the forensic field, highlighting its utility in identification of forensic relevant body fluids, and organ tissues.

Our paper explored a wider number of miRNAs applications in the forensic research such as body fluids identification, wound vitality, MZ twins’ identification and drowning. Since miRNAs exhibit high stability, many studies confirmed these biomolecules as valuable biomarkers for PMI estimation in advanced stages. Molecular methods for PMI determination are based on the time dependent degradation of these molecules. Several recent works have reported the recovery of some miRNAs after compromising treatments and environmental changes.

We also evaluated miRNAs as potential biomarkers in the anti-doping field, since it has been demonstrated that the administration of some substances (i.e., EPO) may determine changes in circulating miRNAs levels. Another forensic field of interest was the applicability of miRNA profiling in sepsis: miRNAs could help the forensic pathologists in the assessment of the cause and mechanisms of death in sepsis negligence claims.

Forensic miRNAs research is in constant development, particularly regarding body fluid identification and post-mortem interval estimation, while other fields are still in early stages.

Although miRNA-based applications are very promising, some difficulties remain to be addressed.

As outlined above, an important limitation is represented by the discrepancies between various studies in different forensic fields. This issue, probably related to differences in the study design (i.e., sampling mode, methods for miRNA extraction and quantification), could be resolved through the use of standardized experimental protocols from inter-laboratory evaluation trials.

First, a miRNA-experimental protocol for forensic research should provide a good quantity and quality of miRNAs from the recovery phase. Forensic samples often contain minimal traces, and techniques that require high amounts of sample are not ideal because it is important to preserve as much material as possible for further DNA profiling. Currently, commercially available miRNA extraction kits are not designed for forensic purposes, but a recent work described the ability to detect miRNAs in DNA extracts obtained using a routinely forensic extraction kit (Qiamp DNA Investigator- Qiagen) [34].

Second, a good miRNA-experimental protocol should adopt qRT-PCR for miRNA expression analysis, as it is actually considered the gold standard approach given its high sensibility and specificity. The use of appropriate internal controls is also encouraged for a proper data validation.

Another relevant topic concerning BFID test (the most explored field in miRNAs potential) is the necessity to establish an accepted set of miRNAs to represent individual body fluid thereby creating a signature of the particular body fluid. A miRNA panel should be validate among different laboratories. However further research is needed to identify a set of unambiguous markers for all forensically relevant body fluids and only few markers has been reported by several groups.

It is still challenging to propose miRNA panels for other forensic applications reviewed in this manuscript, as forensic research is still in its early stages and further investigations are needed to find common miRNA biomolecules that could be adopted as validated biomarkers for courtrooms.

In conclusion, forensic miRNA analysis is still in its infancy and major effort is still required to settle these biomolecules issues as well as to make miRNA quantitation a validated, standardized, and reliable technique for forensic casework routine.

## Figures and Tables

**Table 1 diagnostics-11-00032-t001:** Candidate miRNAs identified in the literature for BFID.

Venous Blood	Menstrual Blood	Saliva	Semen	Vaginal Secretion
miR-451a	miR-214-3p	miR-205-5p	miR-891a	miR-203a
[33,36,39,42,43,44,49,50,58]	[37,43,44,49]	[33,36,39,44,50]	[35,37,41,44,47,50,53,54,58]	[40,41,42]
miR-144-3p	miR-412	miR-203a	miR-888	miR-124a
[35,41,44,58]	[39,50,52]	[36,39,41,58]	[37,38,44,49,53,54,58]	[39]
miR-16	miR-205-5p	miR-658	miR-10b	miR-372
[37,39,58]	[39,43]	[39]	[39,42,50]	[39]
miR-185	miR-203a	miR-142-3p	miR-135b	miR-200c-3p
[35]	[40,43]	[35]	[39]	[40]
miR-126	miR-144-3p	miR-200c	miR-135a	miR-205-5p
[36]	[41,44]	[36]	[35,38]	[40]
miR-150	miR-451a	miR-26b	miR-374	miR-654-5p
[36]	[39]	[47]	[42]	[44]
miR-486	miR-943	miR-223	miR-10a	miR-888
[37]	[42]	[58]	[58]	[44]
miR-203a	miR-1246			miR-155-5p
[41]	[47]			[58]
miR-943	miR-200b			miR-1260b
[42]	[47]			[58]
miR-200b	miR-141-3p			
[47]	[50]			
miR-142-3p				
[50]				

**Table 2 diagnostics-11-00032-t002:** Candidate miRNAs identified in the literature for PMI.

miRNA and PMI				
miRNA	Source	Tissue Type	Stability	Reference
miR- 1-2	Rat	heart	Up to 120 h	[85]
miR-133a, miR-206	mouse	liver	24–48 h	[86]
let-7e, miR-16	Human	bones	1–6 months	[87]
miRNAs as biomarkers for circadian rhythm				
miR-142-5p, miR-541	Human	vitreous humor	low daytime high nighttime	[88]
miR-106b, miR-96	Human	vitreous humor	high daytime low nighttime	[90]
miR-142-5p, miR-219	Human	blood	high daytime low nighttime	[90]
miRNA as internal controls				
miR-203	Rat	skin	Up to 120 h	[91]
miR-125b, miR-143	Rat	spleen	Up to 36 h (25 °C)	[92]
miR-125b, miR-143	Rat	spleen	Up to144 h (4 °C)	[92]
miR-195, miR-200c	Human, rat	lung	Up to 144 h	[93]
miR-1, miR-206	Human, rat	muscle	Up to144 h	[93]
miR-9, miR-125b	Rat	brain	Up to 144 h	[94]
miR-9, miR-125b	Rat	brain	Up to 24 h	[98]
miR-9, miR-125b	Human	brain	Up to 22 h	[99]
miR-1, miR-133a	Human	heart, liver, brain	Up to 180 h	[95]
miR-1, miR-133a	rat	Heart, liver	Up to 180 h	[95]
miR-122, miR-133a	Mouse	heart	Up to 180 h	[96]
miR-122	Mouse	liver	Up to 180 h	[96]
miR-133a	Mouse	skeletal muscle	Up to 180 h	[96]
miR-122, miR-133a	Mouse	heart	Up to 180 h	[97]

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
