# Peer review of "MicroRNAs: An Update of Applications in Forensic Science"

_diagnostics, 2020, doi:10.3390/diagnostics11010032_

Round 1

Reviewer 1 Report

“MicroRNAs: an update of applications in forensic medicine” by Rocchi et al.

The authors did a commendable work and reviewed numerous original articles analyzing miRNAs in different body fluids and tissues in forensic context. A comprehensive review article on forensic applications of microRNA profiling is lacking. The authors considerably improved the presentation of the manuscript. However, there are few more aspects of the presentation, which should be elaborated, in order to achieve that this review represents a straightforward “state-of-the art” snapshot of the field. Moreover, few recently published articles in the field were not considered.

From the reader’s point of view, the manuscript lacks clear statements and conclusion remarks regarding the role of the miRNA in different field of forensic biology. For example, the subheading “2. miRNA and body fluid identification” is extremely hard to follow. The reader would profit from an intelligible list i.e. table with best potential miRNA markers for identification of a body fluid. In the presented Table 1 were markers mentioned several times (since investigated in more studies), what is confusing. The authors might present the miRNA markers in order how they perceived their value based on the quality of original data.

The authors wrote “Finally, it clearly emerges that the use of a miRNAs panel obtained from the combination of different miRNAs, could improve the identification of a particular body fluid.” (Line 170). It would be advantageous to find a suggestion of miRNA panels for identification of different body fluids in completed article. Also, it would be advantageous to find a suggestion of miRNAs (panels) for every reviewed application of miRNA in the forensic biology – e.g., presented as a list with primary sources (Line 512).

The authors wrote: “As outlined above, an important limitation is represented by the discrepancies between findings of various studies. This issue, probably related to differences in study design (i.e. sampling mode, methods for miRNA extraction and quantification), could be resolved through the use of standardized experimental protocols, also at the international scientific community level.” (Line 509). It would be advantageous to find a suggestion of miRNA-experimental protocol.

I find the subheading 7. (“miRNAs and discrimination of monozygotic twins”) needs to be rewritten. Paragraph “In the last few years (Line 371)” seems to be out of scope of the article, or too long - at minimum. The referring to the results of study from Xiao et al. (Line 396) could be summarized with the concluding sentence in Line 403.

The subheading 8. (“miRNAs and anti-doping”) is fuzzy and with several far-fetched conclusions. It should be clearly stated at the beginning of the subheading, which doping substances/procedures are potential targets of miRNA profiling. For example: “This result suggests that miR-122 could be used as new biomarker to detect testosterone abuse [cxvi ].” and state in the next sentence: “Contrasting results were instead obtained in another study…”.

Regarding the subheading 9. (“miRNAs and sepsis”) – was any miRNA marker tested in forensic context?

Please, follow the Instructions for Authors regarding the references.

Please, place the reference numbers by the first mention of the results from other studies. When referring the results of “many studies” (Line 83) or “Most of the studies” (Line 263), please, include the appropriate references.

The authors should take great care in cutting clutter out of many sentences. For example: “potential useful tools” (Line 20) – vs. not useful (?); “their intrinsically small size” (Line 21) – vs. extrinsically (?); “useful internal reference” (Line 331);

Please, take care of correctness of the statements and logic of the sentence (e.g., RNA is co-extracted with DNA, same as miRNA (Line 90); “results in this search failed” (Line 93);

The authors should explain the terms at first mention and be concise in using abbreviations. Some explanations and abbreviations might be omitted, if (since) they do not contribute to the clarity and readability of the text (e.g., Lines 185 -200). By my opinion are some of these parts out of scope of interest of a forensic reader.

Not few awkward terms were used and there are several typing errors in the manuscript (e.g., not limited to “rule” – Line 352). The English language has to be improved.

Please, consider the following suggestions and concerns:

- writing “forensic biology or science” instead “forensic medicine” in the title, since the former represents the subject better.

- to offer a balanced view of the topics and writing style (e.g., “miRNA assays have been pervading forensic medicine and pathology protocols” - Line 61; “In recent years miRNA profiling has been widely used in forensic sciences” – Line 493; “expression levels of let-7e and miR-16 can be used for PMI determination for several months in bone tissue” - Line 340, a suggestion based only on one study based on 71 bones; “drowning murine models […]  miRNA-706 could thus be suggested as a potential marker in forensic investigations to discriminate between SW and FW drowning” - Line 357-365 - suggesting a miRNA marker tested in one animal study).

- introducing a new paragraph, when referring to the results from one study (e.g., Line 110, 130, 136, 269, etc.),

- spell out all numbers beginning a sentence and consider spelling out the numbers zero through nine and using numerals thereafter,

- offering a complete information from the primary data (e.g., Line 282: higher or lower),

- explaining the importance of internal controls,

- explaining and discussing the differences in discordant studies,

- explaining the difference between the peripheral and venous blood (e.g., Line 133).

Author Response

Dear reviewer,

we are grateful for all your recommendations.

We tried to do our best in following your suggestions and we have exstensively reviewed the manuscript.

“MicroRNAs: an update of applications in forensic medicine” by Rocchi et al.

The authors did a commendable work and reviewed numerous original articles analyzing miRNAs in different body fluids and tissues in forensic context. A comprehensive review article on forensic applications of microRNA profiling is lacking. The authors considerably improved the presentation of the manuscript. However, there are few more aspects of the presentation, which should be elaborated, in order to achieve that this review represents a straightforward “state-of-the art” snapshot of the field.

- Moreover, few recently published articles in the field were not considered.

The new version of the manuscript has been integrated with recent literature.

From the reader’s point of view, the manuscript lacks clear statements and conclusion remarks regarding the role of the miRNA in different field of forensic biology.

- For example, the subheading “2. miRNA and body fluid identification” is extremely hard to follow. The reader would profit from an intelligible list i.e. table with best potential miRNA markers for identification of a body fluid. In the presented Table 1 were markers mentioned several times (since investigated in more studies), what is confusing. The authors might present the miRNA markers in order how they perceived their value based on the quality of original data.

We are grateful to know that our current approach on exposing data via Table 1 is not so clear and requires some rethinking.

To note, we decided to modify the entire section 2 following the reviewer’s suggestions: we added very recent literature and we modified Table 1, reporting the miRNAs identified in literature as potential biomarkers for BFID in five forensic relevant body fluids.

We also mentioned the most relevant miRNAs in BFID at the end of subheading 2, to report the summarized data more clearly than the previous version.

- The authors wrote “Finally, it clearly emerges that the use of a miRNAs panel obtained from the combination of different miRNAs, could improve the identification of a particular body fluid.” (Line 170). It would be advantageous to find a suggestion of miRNA panels for identification of different body fluids in completed article.

- Also, it would be advantageous to find a suggestion of miRNAs (panels) for every reviewed application of miRNA in the forensic biology – e.g., presented as a list with primary sources (Line 512).

In our opinion, it is very difficult to propose miRNA panels for all the forensic applications reviewed in this manuscript. Scientific research is still in its early stages and further investigations are needed to find common miRNA biomolecules to be used as validate biomarkers in courtrooms.

- The authors wrote: “As outlined above, an important limitation is represented by the discrepancies between findings of various studies. This issue, probably related to differences in study design (i.e. sampling mode, methods for miRNA extraction and quantification), could be resolved through the use of standardized experimental protocols, also at the international scientific community level.” (Line 509). It would be advantageous to find a suggestion of miRNA-experimental protocol.

We tried to write some suggestions in the Conclusion remarks.

- I find the subheading 7. (“miRNAs and discrimination of monozygotic twins”) needs to be rewritten.

We found your comment extremely helpful and we have rewritten subheading 7.

- Paragraph “In the last few years (Line 371)” seems to be out of scope of the article, or too long - at minimum. The referring to the results of study from Xiao et al. (Line 396) could be summarized with the concluding sentence in Line 403.

Line 371: We have summarized the paragraph trying to follow the suggestions of Reviewer1.

Line 396-403: we tried to explain better the results of these two studies.

- The subheading 8. (“miRNAs and anti-doping”) is fuzzy and with several far-fetched conclusions. It should be clearly stated at the beginning of the subheading, which doping substances/procedures are potential targets of miRNA profiling. For example: “This result suggests that miR-122 could be used as new biomarker to detect testosterone abuse [cxvi ].” and state in the next sentence: “Contrasting results were instead obtained in another study…”.

We found your comment extremely helpful and we have rewritten subheading 8 following your suggestions.

- Regarding the subheading 9. (“miRNAs and sepsis”) – was any miRNA marker tested in forensic context?

Regarding subheading 9, we decided to cite miRNAs analyzed to improve diagnosis of sepsis because miRNAs detectable in blood of sepsis patients could be used also to solve medico-legal disputes (i.e. when a coroner is called to debate a sepsis negligence claim).  

- Please, follow the Instructions for Authors regarding the references.

We prepared the references using Zotero bibliography software package (as suggested by Diagnostics Instruction for authors) following ACS style rules.

- Please, place the reference numbers by the first mention of the results from other studies. When referring the results of “many studies” (Line 83) or “Most of the studies” (Line 263), please, include the appropriate references.

Appropriate references have been added to the text in Line 83 and in Line 263.

Moreover, Line 263 has been changed as follows: “Studies conducted on animal models reported no correlation between miRNA expression levels and early PMI (within 24 hours) .”

- The authors should take great care in cutting clutter out of many sentences. For example: “potential useful tools” (Line 20) – vs. not useful (?); “their intrinsically small size” (Line 21) – vs. extrinsically (?); “useful internal reference” (Line 331);

Line 20:potential useful tools” has been replaced with "promising biomarkers";

Line 21: “intrinsically” has been removed to make the sentence clearer than the previous version;

Line 331:useful internal reference” has been replaced with "appropriate control markers ".

- Please, take care of correctness of the statements and logic of the sentence (e.g., RNA is co-extracted with DNA, same as miRNA (Line 90); “results in this search failed” (Line 93);

Line 90: the sentence has been rewritten as follows: “Moreover miRNAs, as well as mRNAs, can be co-extracted with DNA using common DNA extraction methods. This allows the parallel analysis of DNA profiling and BFID test, very helpful in case of degraded samples or a small amount of forensic traces”.

Line 93: the sentence has been rewritten as follows: “Since numerous published studies in the biomedical field reported that some miRNAs were expressed in a tissue-specific manner, at first, forensic researchers aimed to identify miRNAs specific for one body fluid (present in one body fluid but completely absent in the other body fluids).”

- The authors should explain the terms at first mention and be concise in using abbreviations. Some explanations and abbreviations might be omitted, if (since) they do not contribute to the clarity and readability of the text (e.g., Lines 185 -200). By my opinion are some of these parts out of scope of interest of a forensic reader.

A more concise version of Lines 185-200 has been rewritten as follows: “Also miRNAs have been studied as possible biomarkers for this issue because previously published reports demonstrated a key role of specific miRNAs in the regulation of the wound-healing process.

In particular: miR-21, miR-146a/b miR-142 and miR-155 are characteristic of the inflammatory phase, miR-21  and miR-132  are involved both in inflammatory and proliferative phase, miR-31  and miR-99 [73] are characteristic of the proliferative phase while miR-29a,b,c and miR-192 are involved in remodelling phase.”

- Not few awkward terms were used and there are several typing errors in the manuscript (e.g., not limited to “rule” – Line 352).

Checked and corrected as requested.

- The English language has to be improved.

We tried to do our best in checking the English language.

Please, consider the following suggestions and concerns:

- writing “forensic biology or science” instead “forensic medicine” in the title, since the former represents the subject better.

The title has been modified in “forensic science” as suggested by rewiewer1.

- to offer a balanced view of the topics and writing style (e.g., “miRNA assays have been pervading forensic medicine and pathology protocols” - Line 61; “In recent years miRNA profiling has been widely used in forensic sciences” – Line 493; “expression levels of let-7e and miR-16 can be used for PMI determination for several months in bone tissue” - Line 340, a suggestion based only on one study based on 71 bones; “drowning murine models […]  miRNA-706 could thus be suggested as a potential marker in forensic investigations to discriminate between SW and FW drowning” - Line 357-365 - suggesting a miRNA marker tested in one animal study).

Line 61: This sentence has been substituted with “In recent years miRNA profiling has been studied in various fields of forensic science”;

Line 493: This sentence has been rewritten as follows “In recent years, miRNA profiling acquired an increasing interest among clinicians and researchers, both in diagnostic and therapeutic fields, and it has been also explored in the forensic field”;

Line 340: This sentence has been rewritten as follows “The authors suggested that let-7e and miR-16 could be candidate markers to estimate PMI of several months in bone tissues. Although results from Na et al. are promising, further research is required to consolidate these data with a bigger cohort of long-term PMI samples. Finally, we can conclude that no correlation between early PMI (within 24 hours) and miRNA expression levels have been found but more encouraging results come from the evaluation of miRNAs in late PMI. To note it is necessary to increase the number of studies conducted on human samples. At present, a multi-method approach (evaluation of algor mortis, livor mortis, rigor mortis, biochemical changes in the vitreous humor, and entomological approach) is recommended for an accurate PMI determination and miRNA profiling could represent an ideal tool to supplement the other classical biochemical and molecular techniques.”;

Line 357-365: This paragraph has been rewritten as follows “To the best of our knowledge, only one study conducted by Yu et al. analyzed miRNA expression profiles in an experimental drowning animal model. The experiments were conducted on 13 mice that were killed as follows: three controls were sacrificed by cervical dislocation, five mice were drowned in freshwater (FW), five mice were drowned in saltwater (SW). Differences between miRNA expression profiles were evaluated in brain samples of SW and FW drowning mice by microarray analysis. A total of 158 miRNAs were found to be differentially expressed but only eight of them (miR-6394, miR-706, miR-30c-1-3p, miR-6238, miR-494-3p, miR-669h-3p, miR-135a-1-3p, miR-5109) were significantly upregulated in FW and downregulated in SW drowning compared to controls. By analyzing the target genes of the differently expressed miRNAs only the non-selective cation channel HCN1, which is mainly expressed in the mouse brain and cardiac cells, was predicted to be one of miRNA-706 targets. The authors noticed that HCN1 mRNA was less expressed in FW and overexpressed in SW samples and suggested miRNA-706 as a candidate biomarker in forensic investigations to discriminate between SW and FW drowning. However, this is only a preliminary study, and further investigations with a larger sample cohort are needed to consolidate the author's findings.”

- introducing a new paragraph, when referring to the results from one study (e.g., Line 110, 130, 136, 269, etc.),

Checked and corrected as requested.

- spell out all numbers beginning a sentence and consider spelling out the numbers zero through nine and using numerals thereafter,

Checked and corrected as requested.

- offering a complete information from the primary data (e.g., Line 282: higher or lower),

Line 282: primary data has been specified as follows in the manuscript “Four miRNAs showed significant differential expression between individual died at day and at night, miR-106b and miR-96 in vitreous humour while mir-142-5p and mir-219 in blood (miR-106b and miR-96 levels were respectively higher in daytime deaths and lower in night-time deaths; mir-142-5p and mir-219 levels were higher in daytime deaths and lower in night-time deaths)”.

- explaining the importance of internal controls,

We have explained the importance of internal controls in qRT-PCR analysis in the introduction chapter of the manuscript as follows: “To reduce technical variations introduced during the experimental procedure, target miRNA expression is normalized to a stable internal control, measured in the same sample at the same time. An ideal internal control should be expressed at a constant level, not dependent on biological variations, with length and expression range similar to the target miRNA.”

- explaining and discussing the differences in discordant studies,

We discussed this issue in each paragraph of the manuscript.

- explaining the difference between the peripheral and venous blood (e.g., Line 133).

Peripheral blood, cited at line 133, is referred to the blood collected with venipuncture. The same meaning is attributed to the term “venous blood”.

For this reason, we have substituted the term “peripheral blood” with “venous blood” or the word “blood” alone in the text.  Also Table 1 reports “venous blood”.

Reviewer 2 Report

Line 333: Please clarify the meaning of 'Less than other RNA markers the stability of miRNAs.'

Author Response

Dear reviewer,

Thank you for your suggestion,

We tried to do our best in following your suggestions

Reviewer #2:

- Line 333: Please clarify the meaning of 'Less than other RNA markers the stability of miRNAs.'

Thank you very much for your suggestion. We tried to explain better this concept, re-writing the sentence as follows:

Line 333: “MiRNAs are highly stable compared to RNA species due to their small size and protection by protein and/or lipid matrices, but also miRNAs stability is influenced by the type of tissue available, by environmental conditions such as high temperatures and by body condition as putrefaction”.

- English language and style are fine/minor spell check required.

We tried to do our best in checking the English language.

Round 2

Reviewer 1 Report

Review Report

MicroRNAs: an update of applications in forensic science.

The authors present a comprehensive, considerably improved manuscript. I would like to thank the authors for accepting my suggestions.

However, there are few more aspects of the presentation, which should be elaborated, in order to achieve that this review represents a comprehensible snapshot of the field. Mostly, but not exclusively, it is about structure and style. The text should be better organized and the matter comprehensively described. A large issue is the clutter – it makes the reading so hard, and text unnecessary long.

I find that the authors should keep the following in mind: “Vigorous writing is concise. A sentence should contain no unnecessary words, a paragraph no unnecessary sentences, for the same reason that a drawing should have no unnecessary lines and a machine no unnecessary parts. This requires not that the writer make all his sentences short, or that he avoids all detail and treat his subjects only in outline, but that he makes every word tell.” (Strunk and White, The Elements of Style).

I suggest authors to read the following article (online are available numerous sources with tips how to cut clutter out of the text):

https://www.thoughtco.com/practice-in-cutting-the-clutter-1692770

I recommend that authors severely shorten all sections/headings except the conclusion. The readers would benefit a lot from introducing subheadings in all headings except Introduction and Conclusions. I would advise that at least two longest headings (2. and 4.), have a concluding subheading. Please, correct the numbering of the headings 6-10 (Heading 5 was skipped).

I will try to make my point at two “sample” paragraphs:

Line 99: “Since numerous published studies in the biomedical field reported that some miRNAs were expressed in a tissue-specific manner, at first, forensic researchers aimed to identify miRNAs specific for one body fluid (present in one body fluid but completely absent in the other body fluids). Truly specific miRNAs should directly determine the nature of a body fluid by simply detecting their presence or absence. However, the specificity of most miRNAs did not meet these requirements. because the same miRNA could be found in different body fluids. [85 vs. 50 words]

Line 133: “A further investigation carried out on venous blood and saliva samples, confirmed the findings of previous studies [35,37]. miR-451a was higher in venous blood, as compared to saliva and miR-205 was higher in saliva than in venous blood suggesting that miR-451a and miR-205 could be used to distinguish between venous blood and saliva. Besides the authors highlighted the efficiency of a routinely kit for DNA extraction (QIAamp DNA Mini Kit, Qiagen UK) in providing high yields of microRNA and DNA helpful in case of minute forensic traces [33]. [88 vs. 63 words]

Further note; from the first sentence is unclear who performed “further investigation” – although I find that “Another study” would give enough background information.

Regarding the individual headings:

  1. miRNA and body fluid identification

I am very grateful to the authors for presenting the table 1 at the end of the heading. The table and the following paragraphs (Line 259-270) are real jewels of the heading. The whole heading has approx. 2400 words and that without a single subheading. The heading requires severe shortening: the sentences without clutter are not so oft.

I suggest that authors restructure the heading in a way to present (i) an overview of the subject (NGS and qPCR (e.g., lines 171, 198, 217, 239, 245), extraction, validation, internal controls), (ii) a table with following three paragraphs, and then (iii) further reading about the details of reviewed articles (for readers interested to get in-depth insight of the issue).

I find that such a long heading (although it could be cut to 1600-1800 words) must have subheadings. Currently the heading spreads on approx. 4 pages. Without a structure, the text is really hard to follow.

  1. miRNAs and wound vitality

I suggest that authors introduce the subheading on “exposition to fire” or similar.

  1. miRNAs and time of death determination

This is also a very long, unstructured heading. It is highly recomendable to present an overview of the subject in form of a table. 

  1. miRNAs and discrimination of monozygotic twins

This heading is so long for no reason.

Author Response

Dear reviewer,

Thank you very much for your precious suggestions.

We followed your recommendations point by point.

Review Report

MicroRNAs: an update of applications in forensic science.

The authors present a comprehensive, considerably improved manuscript. I would like to thank the authors for accepting my suggestions. However, there are few more aspects of the presentation, which should be elaborated, in order to achieve that this review represents a comprehensible snapshot of the field. Mostly, but not exclusively, it is about structure and style. The text should be better organized and the matter comprehensively described. A large issue is the clutter – it makes the reading so hard, and text unnecessary long. I find that the authors should keep the following in mind: “Vigorous writing is concise. A sentence should contain no unnecessary words, a paragraph no unnecessary sentences, for the same reason that a drawing should have no unnecessary lines and a machine no unnecessary parts. This requires not that the writer make all his sentences short, or that he avoids all detail and treat his subjects only in outline, but that he makes every word tell.” (Strunk and White, The Elements of Style). I suggest authors to read the following article (online are available numerous sources with tips how to cut clutter out of the text): https://www.thoughtco.com/practice-in-cutting-the-clutter-1692770

Thank you very much for suggesting us this article, it is a great help to improve our writing skills.

I recommend that authors severely shorten all sections/headings except the conclusion. The readers would benefit a lot from introducing subheadings in all headings except Introduction and Conclusions. I would advise that at least two longest headings (2. and 4.), have a concluding subheading.

Please, correct the numbering of the headings 6-10 (Heading 5 was skipped).

We corrected the numbering of headings 6-10 replacing them with the correct numbers 5-9.

I will try to make my point at two “sample” paragraphs:

Line 99: “Since numerous published studies in the biomedical field reported that some miRNAs were expressed in a tissue-specific manner, at first, forensic researchers aimed to identify miRNAs specific for one body fluid (present in one body fluid but completely absent in the other body fluids). Truly specific miRNAs should directly determine the nature of a body fluid by simply detecting their presence or absence. However, the specificity of most miRNAs did not meet these requirements. because the same miRNA could be found in different body fluids. [85 vs. 50 words]

Line99: We have followed your suggestions and we have corrected this line.

Line 133: “A further investigation carried out on venous blood and saliva samples, confirmed the findings of previous studies [35,37]. miR-451a was higher in venous blood, as compared to saliva and miR-205 was higher in saliva than in venous blood suggesting that miR-451a and miR-205 could be used to distinguish between venous blood and saliva. Besides the authors highlighted the efficiency of a routinely kit for DNA extraction (QIAamp DNA Mini Kit, Qiagen UK) in providing high yields of microRNA and DNA helpful in case of minute forensic traces [33]. [88 vs. 63 words]

Line 133: We have followed your suggestions and we have corrected this line.

Further note; from the first sentence is unclear who performed “further investigation” – although I find that “Another study” would give enough background information.

We followed your suggestion and we substituted “further investigation” with “Another work” as follows: "Another work on venous blood and saliva samples confirmed the findings of previous studies: miR-451a was higher in venous blood and miR-205 was higher in saliva".

Regarding the individual headings:

  1. miRNA and body fluid identification

I am very grateful to the authors for presenting the table 1 at the end of the heading. The table and the following paragraphs (Line 259-270) are real jewels of the heading. The whole heading has approx. 2400 words and that without a single subheading. The heading requires severe shortening: the sentences without clutter are not so oft.

I suggest that authors restructure the heading in a way to present (i) an overview of the subject (NGS and qPCR (e.g., lines 171, 198, 217, 239, 245), extraction, validation, internal controls), (ii) a table with following three paragraphs, and then (iii) further reading about the details of reviewed articles (for readers interested to get in-depth insight of the issue).

I find that such a long heading (although it could be cut to 1600-1800 words) must have subheadings. Currently the heading spreads on approx. 4 pages. Without a structure, the text is really hard to follow.

We tried to follow your suggestions, shortening the paragraph to a total of approximately 2000  words.

We performed a reorganization of the entire paragraph, giving a general overview of the subject at the beginning and then reporting the detailed reviewed articles. We hope that also without subheadings the new work is clearer than the previous version.

  1. miRNAs and wound vitality

I suggest that authors introduce the subheading on “exposition to fire” or similar.

We followed your suggestion and we introduced the "exposition to fire " subheading.

  1. miRNAs and time of death determination

This is also a very long, unstructured heading. It is highly recomendable to present an overview of the subject in form of a table. 

We tried to follow your suggestions, shortening the paragraph

We performed a reorganization of the entire paragraph and we introduced a table to summarize the results of the studies.

  1. miRNAs and discrimination of monozygotic twins

This heading is so long for no reason.

We tried to follow your suggestions, shortening the paragraph

This manuscript is a resubmission of an earlier submission. The following is a list of the peer review reports and author responses from that submission.

Round 1

Reviewer 1 Report

The range of PMI during which miRNA can be reliably used for the diagnosis should be mentioned.

The conditions of the body that affects the stability of miRNA should also be mentioned.

Line 24: 'potential useful tools' should be 'potentially useful tools.'

Line 284:  Comma is needed between liver and myocardium.

Lines 248 through 305: Regarding miRNAs and time of death determination, the range of PMI in which miRNA is useful for determination should be mentioned.

Reviewer 2 Report

MicroRNAs: an update of applications in forensic medicine” by Rocchi et al.

A brief summary

The authors reviewed the literature regarding the application of miRNA in the field of forensic science. The review article covers numerous original articles analyzing expression of miRNA in different body fluids and tissues.

General comments 

A comprehensive review article on forensic applications of microRNA profiling is lacking. Two reviews on the potential applications of microRNA profiling in the forensic investigations were published in 2020. Glynn published a (mini-)review “Potential applications of microRNA profiling to forensic investigations” and Yang et al. a review “Research Progress on microRNA in Forensic Medicine as Molecular Markers” in Chinese. In comparison to Glynn, Rocchi et al. provided a much broader insight in the field. However, the article failed to synthesize diverse results and to give other researchers a straightforward “state-of-the art” snapshot of the field. Several aspects of the study and the manuscript should be elaborated.

Some general concerns and suggestions are as follows:

- explain the terms at first mention and be concise in using abbreviations and terms (i.e. Line 138 – biological fluids vs. body fluids vs. body-fluids; MZT vs. MZ twins; L. 198-201; mmu),

- keep the language simple (L. 448/449) and write shorter sentences,

- include the reference as soon as the results from some article are recalled,

- take care of the syntax and clarity of the text (i.e. Line 93 – compared with?; L. 185 – forensic...forensic; L. 312 – “dilemma” implies two alternatives),

- avoid use of pronoun “they and their” where possible, since it is sometimes hard to follow the sentence (i.e. L. 116; L. 434),

- not few sentences are incomplete and fuzzy (i.e. Line 125 – others – which?; L. 222 - compared to?; L. 234 – compared to?; L. 246 – which study? Which miRNAs?; L. 284 – human liver myocardium; L- 287 – reference controls; L. 300 advanced stages of what?, how accurate?; L. 420 - “biomarker signature”; L. 423 - “blood bag processing”, meaning?; L. 440/442 – compared to?),

- avoid use of term “significant”, if not directly related with results of a statistical analyses (i.e. L. 380; L. 427),

- be cautious when using the terms “up- and down-regulation” when recalling the results from a single analysis, since these terms imply temporal relation/trend and “reference” tissue/sample mostly lacks - higher or lower expression of a miRNA might be more appropriate when comparing different body fluids and tissues,

- take care about differences between the British and American English,

- several paragraphs are extremely (or even impossible) to follow and should be rewritten (i.e. L. 117-138; L. 189-91; L. 206-210; L. 353 – 364; L – 365 – 382; L. 392 – 396, model organism, sample size?; L. 425 – 450, a pathognomonic macroscopic (and histologic) finding of a sepsis will “never” be discovered; L-452 – 476),

- not few awkward terms were used (i.e. L. 167 – infertile samples; L. 187 – large harvest; L. 224 – deceased; L. 314 – drowning patterns; L. 323 – strongly statistically deregulated; L. 332 - of the two members of the couple; L. - 381 – nullified; L. 425 – highly orchestrated syndrome; L. 434 – deep integration, septic disease; L. - 437/438; L. 466 – complex pathophysiological dysfunctions),

- avid unclear terms like: normal (controls), deregulated, L. 355 – healthy MZTs vs.?, L. 384 - “suitable for easily accessible material”),

- take care of correctness of the statements (i.e. L. 297/298 – all these are not genes; L. 431 - “miRNA are strictly…”),

- avoid repeating many times how miRNAs are stable, have low molecular weight, etc.,

- in concluding remarks are toxicological aspects mentioned, and the article lacks this issue,

- reference style is not consistent and there are several incomplete references,

- Table 1 is not self-understandable (i.e. meaning of the colors).

Reviewer 3 Report

Interesting paper. Moderate English changes are required.